# Selected Indigenous *Saccharomyces cerevisiae* Strains as Profitable Strategy to Preserve Typical Traits of Primitivo Wine

**Angela Capece**, **Rocchina Pietrafesa \***, **Gabriella Siesto**, **Rossana Romaniello**, **Nicola Condelli** and **Patrizia Romano**

Scuola di Scienze Agrarie, Forestali, Alimentari ed Ambientali, Università degli Studi della Basilicata, 85100 Potenza, Italy; angela.capece@unibas.it (A.C.); gasiesto1@virgilio.it (G.S.); rossanaromaniello@yahoo.it (R.R.); nicola.condelli@unibas.it (N.C.); patrizia.romano@unibas.it (P.R.)

\* Correspondence: rocchina.pietrafesa@unibas.it; Tel.: +39-0971-205585

**Abstract:** Wine production by inoculated fermentation with commercial *Saccharomyces cerevisiae* strains is an ordinary practice in modern winemaking in order to assure the final quality of wine, although this procedure results in the production of highly homogeneous wines. The use of indigenous selected starters represents a useful tool to control alcoholic grape must fermentation, safeguarding the typical sensory characteristics of wine produced from specific regions. In this study, we selected three indigenous *S. cerevisiae* strains among 16 indigenous strains previously isolated from the spontaneous fermentation of Primitivo grapes, which were collected from the vineyards of three different cellars. The three selected starters (one for each cellar) were tested during fermentations at pilot scale by performing in each cellar two trials: one with an indigenous starter (specific for the winery), and one with the commercial starter AWRI796 (common to all the cellars). Starter dominance ability and influence on aromatic quality of the wine were used as criteria to test the suitability of these indigenous starters to be used at the cellar scale. The results obtained in this study showed that the indigenous strains were characterized by very high dominance ability, and the aromatic quality of wine was strongly influenced both by the inoculated strain and the interaction strain/grape must.

**Keywords:** *Saccharomyces cerevisiae*; indigenous strains; dominance level; wine aroma

## 1. Introduction

In the traditional transformation of grape must into wine, the process is carried out by the metabolic activity of various yeast species and genera present on grape skin, in musts, and in winery equipment developing simultaneously or sequentially. Nowadays, the wine fermentation practices widely include the use of commercial *Saccharomyces cerevisiae* starters to ensure the reproducibility of fermentations from year to year and contribute to the production of more balanced wines [1–3]. However, the oenological practice of inoculated fermentation has determined a certain reduction and flattening of sensory characteristics of the final product, as a consequence of the decrease of diversity in microbial populations involved in fermentation [4,5]. Another problem is that the same commercial starter cultures were used for producing different wines, determining a uniformity in the wine characteristics, but the real number of commercial yeast strains is lower than we think, as manufacturers of different brands often designate the same strain with different codes or names [6,7].

One of the main criticisms against commercial starter cultures is that the use of a few commercial selected strains to ferment wines of different variety and origin reduces the uniqueness of the wine bouquet, which is a characteristic of the different vitivinicultural regions throughout the world [8,9].

In fact, the commercial selected yeasts, although possessing the best oenological characters, are not always able to fully develop the typical flavors and aromas of wines coming from different cultivars and produced in a specific region [10,11]. For this reason, spontaneous fermentation, despite the well-known unpredictability of its final result and the risk of microbiological contamination during the process, is still widespread, because the large number of yeasts that develop during this process determine the formation of a more complex aroma, since each spontaneous fermentation is characterized by the growth and activity of a specific population of yeasts [12]. However, the main risk of spontaneous vinification is represented by the possibility to obtain wines with variable qualitative profiles, depending by the evolution of natural microflora in the musts, which changes depending on many factors, resulting in non-reproducible products. In this context, there is a growing interest in the isolation and selection of native yeasts, coming from specific wine-growing regions, because they are probably better adapted to environmental conditions [13–16]. These local strains can control the fermentation process and enhance the typical sensory properties of wine correlated to each production area [17–19]. In fact, scientific evidence has demonstrated the key role played by the yeast strains in the production of characteristic aromas [20,21] and the existence of a wide variability in the production of aroma compounds by the yeast strains of the same species. The development of molecular methods addressed to investigate the yeast population dynamics revealed a significant intraspecific biodiversity, in particular among indigenous strains of *S. cerevisiae*. It must be underlined that genetic diversity reflects strain diversity in the expression of technological parameters, such as the influence on the organoleptic quality of wine.

Therefore, in order to satisfy the modern trend in winemaking to control alcoholic grape must fermentation by using selected starter cultures, many research groups have developed a selection program of indigenous strains, mainly belonging to *S. cerevisiae*, to be used as starter cultures.

In our research, we firstly tested 16 indigenous *S. cerevisiae* strains, previously isolated from the spontaneous fermentation of Primitivo variety grapes, and collected from the vineyards of three different cellars located in the Basilicata region (South Italy). These local yeasts were tested for some oenological traits and tested during laboratory-scale fermentation in order to evaluate the fermentative fitness and strain influence on the content of the main secondary compounds affecting wine aroma. On the basis of the results, three strains were chosen (one for each cellar) and tested during fermentation trials at pilot scale in the three cellars. In each winery, two starters were used, the indigenous (specific for the cellar) and the commercial starter AWRI796 (common to all the cellars). In each trial, the dominance ability of the inoculated starters and their influence on the aromatic and sensorial qualities of the wines were evaluated in order to test the suitability of these indigenous starters to be used at the cellar level.

## 2. Materials and Methods

### 2.1. Yeast Strains

In this work, 16 *S. cerevisiae* strains were used. The yeasts, belonging to the Fermentative Yeast Laboratory collection of Basilicata University, were previously isolated from the spontaneous fermentation of grape must (Primitivo variety). The grapes were collected from the vineyards of three different cellars, all of them located in the South Basilicata region (Italy): "Cantine Cerrolongo" (coded as Cellar B), "Marineto" (coded as Cellar P), "Tenuta Marino" (coded as cellar M); vineyards from the last one were grown under organic farming methods. The strains were previously identified as *S. cerevisiae* by PCR–RFLP analysis of internal transcribed spacers [22] and evaluated by amplification of the interdelta region with δ2/δ12 and δ12/δ21 primer pairs [23], following the protocol described by Capece et al. [24]. The *S. cerevisiae* strains showed 16 distinctive interdelta profiles. All strains were grown at 26°C for 24 h on YPD agar medium (1% (*w/v*) yeast extract; 2% (*w/v*) peptone, 2% (*w/v*) glucose, 2% (*w/v*) agar), and maintained at 4°C until further characterization.

### 2.2. Technological Characterization of S. cerevisiae Strains

The 16 *S. cerevisiae* yeasts were submitted to technological characterization assay in order to select strains with oenological potential. Strain resistance to the different antimicrobial compounds that are present in wine, such as sulfur dioxide ($SO_2$), copper sulfate ($CuSO_4$), and ethanol (EtOH), was evaluated. $SO_2$ and EtOH tolerance of the strains was assessed on agarized grape must (pH 3.6) by adding different amounts of $K_2S_2O_5$ (100, 200, and 300 mg $L^{-1}$) and concentration of ethanol ranging from 12% until 16% (% vol/vol), respectively. Copper resistance was tested as strain ability to grow on synthetic complete medium containing different amounts of $CuSO_4$ (100, 200, and 300 µmol $L^{-1}$). The strain resistance to the three compounds was evaluated on the basis of positive growth after 24 h at 26 °C, in comparison with a control (without compound addition in the medium). The degree of resistance of each strain was reported as the minimal dose that allowed its growth. All the tests were carried out in duplicate.

The production level of hydrogen sulfide ($H_2S$) was evaluated by inoculating the yeasts on bismuth-containing indicator medium BiGGY agar (Oxoid, LtD., Basing Stoke, Hampshire, England), and the plates were incubated at 26 °C for 24 h. On this medium, yeast strains develop colonies with color variable as a function of the increasing amounts of hydrogen sulfide produced, ranging from white/cream (no or low production level of $H_2S$) until brown/black (medium-high production level of $H_2S$).

The presence/absence of killer character was tested in the 16 strains, using as reference strains *S. cerevisiae* NCYC 738 (killer strain) and *S. cerevisiae* NCYC 1006 (sensitive strain). The test was carried out on medium composed of malt extract broth (2%), agar (2%), and methylene blue (0.0003%), and buffered at pH 4.6 with 0.1 M of citric acid phosphate buffer. The reference strains were incorporated into the medium at a concentration of about $10^6$ CFU $mL^{-1}$, and the strains were inoculated by a spot on the plates, which were incubated at 26 °C for 48 h. The killer/sensitivity activity was indicated by the growth inhibition of reference strains [25].

### 2.3. Fermentation Trials at Laboratory Scale

The 16 strains were tested in inoculated fermentation at lab scale to determine strain fermentative performances and the production level of volatile compounds. Fermentations were carried out in 130-mL Erlenmeyer flasks filled with 100 mL of natural red grape must (pH 3.6; sugars 225 g $L^{-1}$; assimilable nitrogen 234 mg $L^{-1}$) supplemented with 50 mg $L^{-1}$ free $SO_2$. Flasks were aseptically inoculated with $10^7$ cells $mL^{-1}$ of each strain pre-culture, grown for 24 h in the same must, and the fermentation was performed at 26 °C. The progress of the fermentative process was monitored by measuring daily weight loss, caused by carbon dioxide ($CO_2$) release during the fermentation. Fermentations were considered to be finished when the weight loss of the samples was constant for three consecutive days. At this time, the experimental wines were refrigerated at 4 °C to allow clarification and stored at −20 °C until required for analysis. All the fermentations were carried out in triplicate.

Experimental wines obtained from inoculated fermentation were analyzed for conventional chemical parameters, such as total and volatile acidity, residual sugars, alcohol, and pH, by a Fourier Transfer Infrared WineScan instrument (FOSS, Hillerød, Denmark). The content of the main secondary compounds influencing wine aroma, such as acetaldehyde, ethyl acetate, acetoin, and higher alcohols (*n*-propanol, isobutanol, and amyl alcohols), were determined by the direct injection of wine samples into a glass column packed with 80/120 Carbopak B/5% Carbowax 20M with an Agilent 7890A gas–chromatograph, as reported in Capece et al. [26].

### 2.4. Pilot-Scale Vinification in Cellar

Based on the results previously obtained, three strains (BP2-33 isolated from B grapes, PP1-13 isolated from grapes collected in P vineyards, and Mpr2-18 isolated from M grapes) were selected to

be used as starter cultures for vinification trials in the three different wineries, in comparison to the commercial starter culture usually utilized in the cellars (AWRI796, MAURIVIN). In each winery, two trials were performed in parallel: a pure culture fermentation with the selected indigenous strain of the corresponding cellar, and a pure culture fermentation with the commercial starter AWRI796. Each cellar used its grape must of Primitivo variety. Indigenous selected strains and the commercial yeast were grown in YPD broth at 26 °C for 2 days with shaking. Fresh cells were collected by centrifugation at $4816\times g$ for 10 min and washed twice in sterile saline solution. For each strain, an aliquot of the cell biomass was inoculated in 500 L of sulfited (50 mg $L^{-1}$) grape must in order to reach an initial concentration of $10^7$ yeast cells $mL^{-1}$ in each fermentation. The fermentation course was monitored daily by determining sugar consumption, and the process was considered to be completed when the residual sugars concentration was less than 2 g $L^{-1}$. Grape must and final wines were analyzed for the chemical parameters by WineScan, as previously reported.

## 2.5. Monitoring of Starter Dominance Ability

In order to evaluate the dominance ability of each starter, during the vinification process at different fermentation stages (beginning, middle, end of the process), samples of Primitivo must/wine were collected. Aliquots of 10-fold dilution of the samples were spread onto Wallerstein Laboratory Nutrient Agar medium (WL Oxoid), and the colony counting was performed after 5 days of incubation at 26 °C.

After viable yeast counting, from each sample and each fermentation phase, 30 colonies showing *Saccharomyces* morphology and 10–20 colonies representative of non-*Saccharomyces* morphology were randomly selected. Species identification was performed by the amplification of the ITS region, followed by restriction analysis using endonucleases *Cfo*I, *Hinf*I, *Hae*III, and *Dde*I [27]. The isolates identified as *S. cerevisiae* were characterized by using inter-δ analysis with aδ2/δ12 primer pair in order to monitor the starter dominance level during the fermentation processes, following the procedure described by Capece et al. [24]. The dominance level of the indigenous and commercial starters in the fermentations was estimated by comparing the δ2/δ12 profile of the inoculated starter with those of the isolates from each fermentation. *S. cerevisiae* strains that were different from the inoculated starters were submitted to technological characterization, following the protocol previously described.

## 2.6. Volatile Composition of Experimental Wines

The content of the major volatile compounds of the final wines were analyzed by a GC glass column packed, as previously reported. Other aromatic compounds, such as acetates and ethyl esters, volatile fatty acids, and terpenes were analyzed by SPME-GC-MS in splitless mode with a DB-WAXTER (Agilent) column, following the procedure described by López-Martínez et al. [28].

## 2.7. Sensory Evaluations

In order to assess the liking scores of samples, a hedonic test was conducted. Eighty-four regular consumers of wine participated in the experiment. A nine-point hedonic scale ranging from "dislike extremely" (1) to "like extremely" (9) was used. Samples were served in completely randomized and balanced order among subjects and evaluated at service temperature (16 ± 2 °C). For each sample, subjects received a glass containing 15 mL of wine. The presentation order of samples was balanced for first-order and carry-over effects. Subjects were asked to observe, smell, and taste the samples, and to report their overall liking score. Between the evaluation of two samples, subjects were asked to rinse their mouths with water for 30 s, to eat a crackers for 30 s, and finally to rinse their mouths with water for a further 45 s. The evaluations were performed in individual booths. The data were collected using the FIZZ computer system (Biosystemes, Couternon, France).

*2.8. Data Analysis*

Each test was carried out independently in duplicate/triplicate, and the results are represented as the average with the corresponding standard deviation (±SD). Levels of secondary compounds and chemical parameters detected in wines from laboratory and pilot-scale fermentations were submitted to statistical analysis by one-way analysis of variance (ANOVA); the statistical significance was set at $p \leq 0.05$. Tukey's test was used to compare the mean values of secondary compounds between fermentation by indigenous strains isolated from the same cellar. Canonical variate analysis (CVA) was carried out to compare the secondary compounds and the main chemical parameters detected in wines by 16 indigenous strains during lab-scale fermentations. Principal component analysis (PCA) was carried out on products of alcoholic fermentation in the three cellars at pilot-scale vinifications. The PAST software ver. 1.90 [29] was used for all the statistical analyses. Regarding sensory evaluations of pilot-scale fermentations, liking scores from six wines were analyzed by using a one-way ANOVA model in order to estimate the sample effect. Significant differences between liking scores were tested by means of the least significant difference (L.S.D.) post hoc test at a level of significance of 95%.

## 3. Results and Discussion

*3.1. Technological Characterization of S. cerevisiae Strains*

In the first phase of the work, 16 *S. cerevisiae* strains previously isolated from the spontaneous fermentation of grapes were studied for some technological characteristics, such as resistance to antimicrobial compounds (sulfur dioxide, copper, and ethanol) and hydrogen sulfide production (Table 1). All the strains exhibited a high tolerance to ethanol, growing on plates containing 16% *v/v* of EtOH (data not shown), whereas a certain variability was found for $SO_2$ and copper resistance.

**Table 1.** Main technological characteristics of the 16 *S. cerevisiae* strains tested in this study.

| Strain Code | Strain Origin | Killer Activity | $SO_2$ Resistance [a] | Cu Resistance [b] | $H_2S$ Production |
|---|---|---|---|---|---|
| BP1-13 | | + | 300 | 300 | Low |
| BP1-29 | | − | 300 | 200 | Low |
| BP1-33 | Cellar B | − | 300 | 100 | Low |
| BP2-17 | | + | 300 | 300 | Low |
| BP2-33 | | + | 300 | 300 | Low |
| PP1-1 | | + | 300 | 100 | Medium |
| PP1-13 | | + | 300 | 300 | Low |
| PP1-15 | Cellar P | + | 300 | 300 | Low |
| PP1-31 | | + | 300 | 300 | Medium |
| PP2-22 | | + | 100 | 300 | Low |
| Mpr1-24 | | + | <100 | 300 | Medium |
| Mpr2-18 | | + | 100 | 300 | Medium |
| Mpr2-26 | | + | 200 | 300 | Medium |
| Mpr2-28 | Cellar M | − | 300 | 300 | Medium |
| Mpr2-42 | | + | 100 | 300 | Medium |
| Mpr2-43 | | + | 300 | 300 | Low |

[a]: expressed as mg $L^{-1}$ of potassium metabisulfite; [b]: expressed as μmol $L^{-1}$ of copper sulfate.

All the yeasts developing to the highest tested doses of $SO_2$ were isolated from cellar B and P (except one), while a certain variability was found among the yeasts from cellar M. Furthermore, these last strains showed the highest resistance to copper sulfate, as well as the majority of strains isolated from cellar P. This result is probably related to the application of organic farming methods in cellar M. As it is well known, the use of copper sulfate is allowed in organic viticulture, as copper formulates are effective against a high number of crops pests, and this compound is considered a traditional fungicide, as it has been used against powdery mildew since the 1880s. Yeasts use several mechanisms to respond to environmental stresses, and these mechanisms have been shown to contribute to the adaptation of wine yeast genomes [30]. It was reported that the acquisition of resistance to copper sulfate could

be associated with the ancient use of this fungicide in vineyards, and the acquisition of resistance to copper sulfate is one of the domestication-related traits found among *S. cerevisiae* wine strains [31]. Considering that in the last years, the interest of consumers for organic products, and also for wine, is increasing, *S. cerevisiae* strains possessing high copper resistance are very interesting for winemaking. In fact, these starters can tolerate the high level of copper residues present in grape must coming from an organic farming system, assuring a regular evolution of fermentative process [32].

Regarding hydrogen sulfide production, the 16 strains exhibited a medium and low production level of this compound, and the lowest production level was found among strains isolated from the grapes of cellar B. The killer assay revealed that 81% of the studied strains exhibited killer activity against the reference sensitive strain, which is an interesting result, as the use of starter-possessing killer activity may potentially favor the strain dominance during winemaking.

### 3.2. Laboratory Fermentations

The fermentative performance of the 16 *S. cerevisiae* strains was evaluated in inoculated fermentations at laboratory scale. The process lasted 12–15 days; strains isolated from grapes of cellar B exhibited the highest fermentative vigor, since they produced from 1.58 to 2.33 g of $CO_2$ per 100 mL within 48 h, whereas the maximum level of $CO_2$ per 100 mL produced after 48 h was 1.67 and 1.96 among strains isolated from samples of cellar P and M, respectively (Table 2).

**Table 2.** Oenological characteristics (average and standard deviation of three independent experiments) of 16 different *S. cerevisiae* strains. For each parameter, superscript letters mean significant differences at $p \leq 0.05$ among starters isolated from the same grape must (i.e., strains coded with BP).

| Strain | Ethanol * | Fructose ** | Glucose ** | Total Acidity ** | Volatile Acidity ** | FV |
|---|---|---|---|---|---|---|
| BP1-13 | 11.10 ± 0.14 [a] | 0.90 ± 0.14 | 0.83 ± 0.11 | 9.72 ± 0.13 [a] | 0.58 ± 0.03 [a] | 1.59 ± 0.01 [a] |
| BP1-29 | 10.66 ± 0.12 [b] | 0.83 ± 0.07 | 0.83 ± 0.07 | 8.65 ±0.12 [b] | 0.36 ± 0.05 [b] | 1.89 ± 0.19 [a] |
| BP1-33 | 10.86 ± 0.03 [a,b] | 0.09 ± 0.28 | 1.10 ± 0.14 | 9.30 ± 0.28 [a] | 0.45 ± 0.04 [a,b] | 1.70 ± 0.05 [a] |
| BP2-17 | 11.22 ± 0.01 [a,c] | 1.00 ± 0.28 | 0.84 ± 0.04 | 9.84 ± 0.04 [a] | 0.59 ± 0.04 [a] | 1.58 ± 0.01 [a] |
| BP2-33 | 11.24 ± 0.01 [a,c] | 1.10 ± 0.01 | 0.83 ± 0.07 | 9.72 ± 0.13 [a] | 0.59 ± 0.04 [a] | 2.33 ± 0.30 [a,b] |
| PP1-1 | 11.24 ± 0.04 | 1.14 ± 0.06 [a] | 1.10 ± 0.28 | 9.44 ± 0.06 [a] | 0.42 ± 0.04 | 1.48 ± 0.04 |
| PP1-13 | 10.97 ± 0.03 | 1.10 ± 0.14 [a] | 1.10 ± 0.11 | 9.84 ± 0.2 [a,b] | 0.45 ± 0.01 | 1.62 ± 0.22 |
| PP1-15 | 11.10 ± 0.13 | 1.34 ± 0.07 [a,c] | 1.24 ± 0.07 | 10.09 ± 0.14 [b] | 0.48 ± 0.01 | 1.47± 0.07 |
| PP1-31 | 11.23 ± 0.24 | 0.90 ± 0.14 [a,b] | 1.22 ± 0.28 | 9.23 ± 0.08 [a,c] | 0.41 ± 0.08 | 1.45 ± 0.02 |
| PP2-22 | 10.92 ±0.03 | 0.50 ± 0.07 [b] | 0.90 ± 0.28 | 8.87 ± 0.18 [a,c] | 0.35 ± 0.06 | 1.67 ± 0.14 |
| Mpr1-24 | 11.39 ± 0.31 [a] | 1.31 ± 0.14 [a,b] | 0.44 ± 0.06 [a] | 6.84 ± 0.16 [a] | 0.28 ± 0.03 [a] | 1.33 ± 0.01 [a] |
| Mpr2-18 | 10.93 ± 0.04 [a] | 0.83 ± 0.07 [a] | 0.73 ± 0.07 [a,b] | 7.44 ± 0.06 [b] | 0.31 ± 0.01 [a] | 1.96 ± 0.30 [a,b] |
| Mpr2-26 | 10.05 ± 0.07 [b] | 1.24 ± 0.07 [a,b] | 1.31 ± 0.28 [b] | 10.44 ± 0.21 [c] | 0.52 ± 0.01 [a,b] | 1.40 ± 0.09 [a] |
| Mpr2-28 | 11.08 ± 0.15 [a] | 0.90 ± 0.14 [a,b] | 1.36 ± 0.14 [b] | 9.24 ± 0.23 [d] | 0.45 ± 0.14 [a,b] | 1.33 ± 0.12 [a,c] |
| Mpr2-42 | 10.95 ± 0.07 [a] | 1.00 ± 0.28 [a,b] | 1.10 ± 0.28 [a,b] | 8.88 ± 0.17 [e] | 0.39 ± 0.03 [a] | 1.49 ± 0.14 [a] |
| Mpr2-43 | 11.27 ± 0.08 [a] | 1.44 ± 0.01 [b] | 0.90 ± 0.14 [a,b] | 10.30 ± 0.14 [f] | 0.67 ± 0.03 [b] | 1.53 ± 0.02 [a] |

\* = %*v/v*; \*\* = g $L^{-1}$, FV = fermentative vigor expressed as g $CO_2$/day, measured at the first two days of the fermentation.

The statistical analysis of the main chemical parameters of the experimental wines, produced by 16 different *S. cerevisiae* strains, showed main variability among wines obtained with strains from the grapes of cellar M than wines produced with strains from the B and P samples (Table 2). All the strains completed the fermentations with a residual sugar content, both as glucose and fructose, lower than 1.5 g $L^{-1}$ (Table 2). As regards the other parameters, such as ethanol and volatile acidity, wines obtained with strains from the P samples showed less variability than experimental wines obtained with strains isolated from grapes of the B and M cellars. However, in all the experimental wines, the volatile acidity ranged from 0.3 to 0.67 g $L^{-1}$, always being within the acceptable limits. It has been reported that the optimal concentration of acetic acid in wine is 0.2–0.7 g $L^{-1}$, and the acceptability level of this parameter is comprised between 0.7–1.1 g $L^{-1}$, depending on the style of wine [33], whereas the

OIV [34] states that the maximum acceptable limit for volatile acidity in most wines is 1.2 g L$^{-1}$ of acetic acid.

Quantitative data from the volatile compounds usually present in high quantity in wines, such as acetaldehyde, *n*-propanol, isobutanol, amyl alcohols, and ethyl acetate, analyzed in the experimental wines obtained with the selected strains, are reported in Table 3.

**Table 3.** Volatile compounds (mg L$^{-1}$) detected in experimental wines produced by 16 *S. cerevisiae* strains during lab-scale fermentation. Data are means ± standard deviation of three independent experiments. For each compound, superscript letters indicate significantly different values (Tukey's test. $p \leq 0.05$) among strains isolated from the same cellar.

| Strains | Acetaldehyde | Ethyl Acetate | *N*-Propanol | Isobutanol | D-Amyl Alcohol | Isoamyl Alcohol |
|---|---|---|---|---|---|---|
| BP1-13 | 61.37 ± 7.83 | 10.27 ± 1.25 [a] | 29.76 ± 0.45 | 39.63 ± 0.08 [a] | 83.23 ± 4.09 [a] | 154.25 ± 49.20 [a] |
| BP1-29 | 41.26 ± 12.28 | 10.68 ± 1.23 [a] | 33.63 ± 4.40 | 36.88 ± 5.48 [a] | 96.78 ± 12.87 [a] | 185.13 ± 31.18 [a] |
| BP1-33 | 56.14 ± 3.44 | 11.05 ± 0.84 [a,b] | 33.87 ± 2.41 | 52.96 ± 3.84 [b,c] | 72.84 ± 3.67 [a,b] | 152.93 ± 13.34 [a] |
| BP2-17 | 52.12 ± 0.86 | 12.80 ± 0.49 [a,b] | 30.76 ± 2.08 | 42.42 ± 2.27 [a,c] | 103.58 ± 7.19 [a,c] | 318.02 ± 5.78 [b] |
| BP2-33 | 70.54 ± 8.60 | 14.35 ± 0.26 [b] | 35.52 ± 1.41 | 47.89 ± 1.40 [a,c] | 94.41 ± 3.67 [a] | 218.62 ± 10.91 [a] |
| PP1-1 | 36.82 ± 2.94 [a] | 11.16 ± 0.31 | 32.55 ± 1.83 | 42.00 ± 3.21 [a] | 90.85 ± 5.72 | 169.26 ± 8.34 [a] |
| PP1-13 | 55.34 ± 0.70 [a] | 10.78 ± 1.77 | 33.29 ± 5.45 | 45.45 ± 8.16 [a] | 84.08 ± 1.75 | 122.56 ± 14.86 [a,b] |
| PP1-15 | 40.57 ± 1.67 [a] | 13.09 ± 0.80 | 36.35 ± 1.00 | 54.34 ± 1.15 [a,b] | 68.14 ± 16.07 | 116.86 ± 25.20 [a,b] |
| PP1-31 | 35.31 ± 3.19 [a] | 11.05 ± 0.08 | 31.13 ± 1.95 | 32.82 ± 0.86 [a,c] | 90.23 ± 11.88 | 206.62 ± 11.41 [a,c] |
| PP2-22 | 197.81 ± 28.99 [b] | 10.31 ± 0.24 | 41.48 ± 3.50 | 39.43 ± 3.32 [a] | 104.12 ± 8.47 | 168.34 ± 14.73 [a] |
| Mpr1-24 | 35.27 ± 3.76 [a] | 12.63 ± 0.13 [a,b] | 39.41 ± 0.39 [a] | 37.73 ± 1.01 | 95.55 ± 5.20 [a] | 238.12 ± 9.48 [a] |
| Mpr2 -18 | 53.22 ± 2.07 [b,c] | 9.81 ± 1.26 [a] | 28.35 ± 3.88 [b] | 52.05 ± 6.19 | 122.67 ± 2.24 [b] | 294.88 ± 10.56 [b] |
| Mpr2-26 | 51.61 ± 0.99 [b,c] | 11.51 ± 0.20 [a] | 32.09 ± 0.54 [a,b] | 42.88 ± 3.65 | 73.49 ± 1.43 [c] | 168.32 ± 2.54 [c] |
| Mpr2-28 | 42.48 ± 0.99 [a,c] | 11.45 ± 0.33 [a] | 37.69 ± 2.22 [a] | 45.64 ± 2.34 | 85.05 ± 2.64 [a,c] | 149.64 ± 1.04 [c] |
| Mpr2-42 | 40.51 ± 7.59 [a,c] | 11.56 ± 1.54 [a] | 37.79 ± 2.96 [a] | 40.24 ± 4.37 | 77.93 ± 2.10 [a,c] | 148.30 ± 22.44 [c] |
| Mpr2-43 | 61.05 ± 1.14 [b] | 15.43 ± 0.62 [b] | 38.21 ± 0.83 [a] | 46.62 ± 2.17 | 95.12 ± 6.04 [a] | 206.49 ± 4.72 [a,c] |

As regards acetaldehyde, only the strain PP2-22 produced too high a level of this compound, whereas the content detected in the other wines was in the usual range (10–75 mg L$^{-1}$). Low acetaldehyde level contributes to fruity flavors, while high concentrations (>200 mg L$^{-1}$) confer "flatness" in wines [35].

The wines produced by strains from M grapes exhibited the highest variability for this compound, whereas similar values of acetaldehyde content were found among samples obtained by strains isolated from B grapes. Generally, the strains from M grapes were characterized by the highest variability for the production level of these secondary compounds. In fact, significant differences were detected among the different fermentations performed by M strains for almost all the compounds, except isobutanol. Among the analyzed compounds, the most variable was the isoamyl alcohol, which was produced at significant different levels in all three groups of experimental wines.

As already reported by other studies [36–38], these results confirmed that from fermentation of the same grape must, different strains of *S. cerevisiae* can produce significantly different amounts of aromatic compounds, as a consequence of both the differential ability of wine yeast strains to release varietal volatile compounds from grape precursors and to synthesize de novo volatile compounds.

A multivariate analysis of variance (MANOVA), based on strain origin as an independent factor, was performed on the data obtained from an analysis of experimental wines, both chemical parameters and aromatic compounds. This analysis showed significant differences between fermentation obtained by strains of different origin (Wilk's lambda 0.0001089, $p < 0.00009776$).

Furthermore, the same data were submitted to canonical variate analysis (CVA), considering the strain origin as an independent factor. This approach allows to visualize the relative position of the different production levels of secondary compounds and chemical parameters of experimental wines in the multivariate statistical space, by maximizing the variation related to the strain origin (Figure 1). The canonical variate analysis of the data demonstrated a clear discrimination between wines in the function of isolation origin of inoculated strains. In fact, wines obtained by inoculating strains from grapes of the same isolation origin were grouped together (i.e., all the wines produced by strains isolated from the P samples were grouped in the left-high panel). The first and second canonical

axes, which accounted for 84.89% and 15.11% respectively of the total variance, discriminate the three groups. This distribution confirms that the production level of secondary compounds and chemical parameters are significantly different in the three groups. The main differences are due to amyl alcohols, acetaldehyde, and volatile acidity. These results confirm that strain origin affects strain metabolic activity, probably as a consequence of the selective pressure exerted by the environment on natural microflora [39]. It was reported that yeast strains develop physiological and metabolic adaptations in response to specific environmental conditions [40], and different authors [41,42] demonstrated the existence of a correlation between strain origin and the characteristics of wines obtained inoculating yeasts strains isolated from different wine regions. In this way, it might be possible to associate specific indigenous strains with a specific region, or with a "terroir", a term that was traditionally associated only with grape variety, climate, and soil. Recent studies [41,43,44] put evidence in on the existence of a microbiological aspect of "terroir", highlighting that wine microbioma (microorganisms influencing both vine growth and wine characteristics) exhibits regional differentiation.

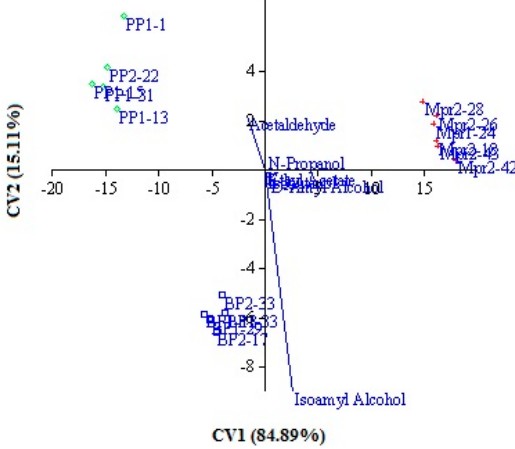

**Figure 1.** Scatter plot of canonical variate analysis on secondary compounds and conventional chemical parameters determined in the experimental wines obtained by inoculating 16 *S. cerevisiae* strains.

### 3.3. Pilot-Scale Vinification: Monitoring of Starter Dominance Ability

On the basis of these results, three indigenous strains, one from each winery, BP2-33, PP1-13 and Mpr2-18, possessing suitable oenological characteristics, such as high resistance to antimicrobial compounds, medium/low $H_2S$ production (Table 1), good fermentative performance (Table 2), and the balanced production of secondary compounds (Table 3), were selected as starters to perform pilot-scale vinification in the three cellars (B, P, and M). These strains were tested in comparison to the commercial strain AWRI796, which is commonly used in the three wineries.

In order to evaluate the capacity to dominate the fermentative process of these three indigenous strains in comparison to the commercial starter, yeast isolation was performed at different fermentation times (beginning, middle, and end) in each vinification trial. All the yeasts isolated were identified by restriction analysis of the ITS region, and all the colonies identified as *S. cerevisiae* were characterized by amplification of the interdelta region (with primer pair δ2/δ12). The strain dominance at the different sampling times, expressed as a percentage of colonies showing the same molecular profiles of the inoculated starter in the six fermentations, was reported in Table 4. All the indigenous starters displayed a dominance capacity higher than that of the commercial strain in all the cellars. In fact, all the colonies isolated from the vessel inoculated with the indigenous starter BP2-33 showed one unique profile throughout the entire fermentation process and corresponding to the inoculated yeast, whereas the indigenous starters tested in the cellars P and M showed a slightly lower dominance level, corresponding to 90.5 (PP1-13) and 96.3% (Mpr2-18) at the end of the process. The commercial starter dominated the fermentation in cellar M, with a dominance of 100% in the final step of the process, and in cellar B, where the dominance ability was about 80% at the end of fermentation, and only two

molecular profiles different from the profile of the inoculated starter were found (Table 4). On the contrary, this starter showed a very low dominance level in cellar P, which was equal to 44% and 63.6% in the middle and final phases of fermentation, respectively. Isolates from a vessel inoculated with the commercial strain in cellar P showed nine different interdelta profiles during the middle stage of fermentation, and three of these profiles (c, e, l, Table 4) were found also in the final phases of the fermentation.

**Table 4.** Dominance level of commercial (AWRI 796) and indigenous *S. cerevisiae* starters during pilot-scale vinifications. The percentage of isolates showing molecular profiles different from starters, indicated with different letters for each cellar, is given in the brackets.

| Cellar | Starter | Fermentation Time | | |
|---|---|---|---|---|
| | | Beginning | Middle * | End |
| B | BP2-33 | 100% | 100% | 100% |
| | AWRI 796 | 100% | 87.5% (4.5% a; 2.0% b 2.0% c; 4.0% d) | 79.3% (13.8% a; 6.9% d) |
| P | PP1-13 | 100% | 96.8% (0.8% a; 2.4% b; 1.6% c) | 90.5% (1.0% a; 6% b; 2.5% c) |
| | AWRI 796 | 100% | 44% (2.7% a; 1.3% b; 5.2% c; 1.3% d; 11.7% e; 6.7% f; 22.7% l; 2.7% m; 1.7% n) | 63.6% (9.0% c; 9.0% e; 18.4% l) |
| M | Mpr2-18 | 100% | 100% | 96.3% (3.7% a) |
| | AWRI 796 | 100% | 96.9% (3.1% a) | 100% |

* = fermentation stage with the reduction of 50% of the sugars.

In this case, other *S. cerevisiae* strains, different from the inoculated starter, participated significantly in the fermentative process. Other authors reported low dominance percentages of commercial yeasts in inoculated winery fermentation [41]. In a study reporting the distribution of wine yeasts in different commercial wineries [45], the authors found a general displacement of the autochthonous strains of *S. cerevisiae* by the commercial strain inoculated, although not in all the wineries analyzed. In fact, in one cellar, the commercial strain, although predominant in the process, allowed the growth of other two strains, whereas in another winery, no imposition of commercial strain was observed. Different reasons were reported to justify the non-implantation of commercial yeast starters, such as the use of lower doses of dry yeast than those recommended, osmotic and oxidative stress, or the use of inappropriate procedures of rehydration [44]. In our study, commercial and indigenous starters were prepared in the laboratory by following the same procedure for all of them; as a consequence, the different dominance level cannot be related to the employed modality. In order to find the characteristics explaining the low dominance level of some starters, *S. cerevisiae* strains different from inoculated starters were studied. In particular, among the yeasts isolated from all the fermentations and showing interdelta profiles different from those of the inoculated starters, a strain representative of each interdelta profile was selected. These strains were tested for resistance to antimicrobial compounds (EtOH, $SO_2$, and copper) and killer activity. All these natural isolates showed variable resistance levels to $SO_2$ and copper, whereas almost all of them showed higher ethanol tolerance than inoculated starters (data not shown). In fact, the maximum dose of ethanol tolerated was 20% *v/v* for all the isolates, whereas the starters were grown until 16% *v/v* of EtOH. As regards the killer activity, variable results were found: some isolates exhibited killer activity against reference strains, whereas others were sensitive to killer protein (data not shown). These results might suggest that the competitiveness of the *S. cerevisiae* population of grape must toward inoculated starters was correlated to high ethanol tolerance. Therefore, indigenous strains were characterized by higher ethanol tolerance than that the inoculated starters developed during the process. However, other traits not investigated in this study might be involved in the high competitiveness of indigenous *S. cerevisiae* strains present in grape must.

As regards the colonies identified as non-*Saccharomyces*, the population distribution of the different species over all the fermentations is reported in Figure 2. All the colonies isolated from fermentations performed in cellars B and P belonged to *Hanseniaspora uvarum* and *Candida stellata*, whereas in the cellar M besides these two species, *C. agrestis* was also found. In cellar B, *H. uvarum* and *C. stellata* were

found both at the beginning and middle phases of fermentations inoculated with commercial strain, and only in the first stage of the process, when the indigenous selected BP2-33 strain was inoculated (Figure 2a). In cellar P, the same non-*Saccharomyces* species was detected only in the first stage of fermentation conducted by both the commercial strain and the indigenous starter PP1-13, whereas *C. stellata* was found also in the middle time of fermentation guided by PP1-13 (Figure 2b). As shown in Figure 2c, in cellar M, at the beginning of the process, *H. uvarum*, *C. stellata*, and *C. agrestis* were isolated from both the inoculated fermentations, whereas in the middle phase of the process, *H. uvarum* and *C. stellata* were present only among colonies isolated from a vessel inoculated with commercial starter, as already reported for the trials performed in cellar B (Figure 2a). This result might indicate that the indigenous starters selected for cellars B and M are highly competitive against non-*Saccharomyces* yeasts compared with the indigenous strain selected for cellar P. Other authors found that the presence of non-*Saccharomyces* species throughout the fermentation process is a consequence of the type of strain used as starter culture. Tofalo et al. [46] reported the presence of non-*Saccharomyces* species throughout all the processes; in this study, an indigenous *S. cerevisiae* strain starter with low alcoholic performance was used as a starter culture, and this could have allowed the growth of non-*Saccharomyces* strains. These findings underline the fundamental role of the screening program followed in the laboratory to select the most promising indigenous *S. cerevisiae* strains able to efficiently ferment the musts.

As expected, these non-*Saccharomyces* yeasts were not found at the end of all the pilot-scale fermentations performed in the three cellars.

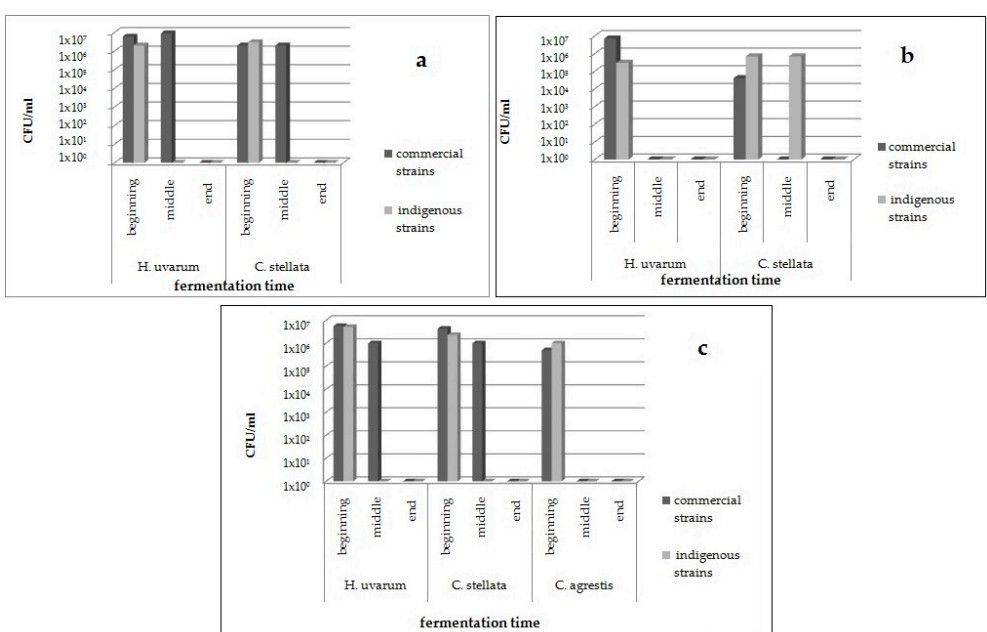

**Figure 2.** Cell counts of indigenous non-*Saccharomyces* yeast species throughout pilot-scale vinification in cellar B (**a**), P (**b**), M (**c**).

### 3.4. Pilot-Scale Vinification: Analysis of Wines

The chemical parameters detected in the wines obtained by the six pilot-scale vinifications, performed in the three different cellars, are shown in the Table 5. The ethanol content ranged between 11.74–15.12 *v/v*, with the highest level in wines obtained in cellar M, both by inoculating indigenous and commercial starters. As a consequence, the wines obtained in this cellar contained the lowest amounts of sugar residuals. All six wines contained a similar level of volatile acidity, which was comprised in the desirable range.

**Table 5.** Chemical parameters (average and standard deviation of two experiments) of the wines obtained in three cellars (B, P, and M) with a commercial *S. cerevisiae* strain (AWRI796) and indigenous *S. cerevisiae* strains isolated from each cellars (cellar B: BP2-33; cellar P: PP1-13; cellar M: Mpr2-18).

| Cellar | Strain | Ethanol [a] | Fructose [b] | Glucose [b] | Total Acidity [b] | Volatile Acidity [b] |
|--------|--------|-------------|--------------|-------------|-------------------|----------------------|
| B | AWRI 796 | 11.74 ± 0.09 | 5.60 ± 0.02 | 5.60 ± 0.05 | 10.80 ± 0.8 | 0.65 ± 0.07 |
|   | BP2-33 | 11.81 ± 0.04 | 4.50 ± 0.06 | 4.80 ± 0.04 | 10.40 ± 0.16 | 0.55 ± 0.13 |
| P | AWRI 796 | 12.07 ± 0.02 | 1.96 ± 0.11 | 5.00 ± 0.11 | 5.80 ± 0.07 | 0.42 ± 0.12 |
|   | PP1-13 | 11.83 ± 0.42 | 1.77 ± 0.14 | 4.40 ± 0.12 | 6.20 ± 0.03 | 0.44 ± 0.07 |
| M | AWRI 796 | 14.40 ± 0.07 | 0.70 ± 0.07 | 1.20 ± 0.09 | 8.60 ± 0.12 | 0.55 ± 0.05 |
|   | Mpr2-18 | 15.12 ± 0.11 | 2.30 ± 0.10 | 1.80 ± 0.07 | 7.50 ± 0.10 | 0.52 ± 0.10 |

[a] = % *v/v*; [b] = g L$^{-1}$.

The aromatic compounds detected in the six wines were reported in Table 6, in which the volatile compounds were subdivided into four chemical classes—namely, esters, alcohols, terpenes, and aldehydes. Higher alcohols represented the most abundant group in the analyzed samples, followed by esters and aldehydes. Fusel alcohols can contribute a positive flavor in the wine when present in concentrations below 300 mg L$^{-1}$, whereas concentrations above 400 mg L$^{-1}$ have a detrimental effect on wine aroma [9]. In all the analyzed wines, the concentration of total fusel alcohols was below 400 mg L$^{-1}$, except for wines produced in cellar M, both by the indigenous and commercial starters (Table 6), and the most abundant alcohol in all the wines was isoamyl alcohol. As regards wines produced in cellar B, the content of higher alcohols in wine produced by the indigenous starter was significantly higher than the level detected in wine produced by AWRI796, except for *n*-propanol content. In addition, for wines produced in cellars M and P, statistically significant differences between wines from indigenous and commercial starters were found for almost all the higher alcohols detected in this study.

The esters are an important group that can significantly affect wine aroma. The fermentation esters associated with wine fruitiness are divided in two groups: acetate esters (mainly: ethyl acetate, 2-phenyl ethyl acetate, 3-methyl-1-butanol acetate or isoamyl acetate, hexyl acetate) and ethyl fatty acid esters. Among the esters, the main wine ester is ethyl acetate, which can contribute a desirable fruit fragrance at concentrations lower than 150 mg L$^{-1}$, whereas it contributes an unpleasant odor at concentrations higher than this value. The highest amounts of this compound were detected in wines produced in cellar B, both by indigenous and commercial starters. However, any pilot-scale wine showed an ethyl acetate content less than 50 mg L$^{-1}$. Other esters present in high concentrations were isoamyl acetate (banana aroma) and ethyl hexanoate (fruity, floral aroma), both produced by all the starters in concentrations higher than their threshold values of 0.03 mg L$^{-1}$ and 0.05 mg L$^{-1}$, respectively [47]. As already reported for higher alcohols, statistically significant differences for the content of all the esters detected in this study were found between wines obtained with commercial and indigenous strains, in particular in cellar M. It was reported [48–50] that ester concentration is affected by different fermentation conditions, such as temperature, aeration, and sugar content, other than yeast strain used. By considering that the same fermentative conditions were used in each cellar, the yeast starters used by the winemakers plays a fundamental role on the ester content of the wines.

As regards aldehydes, it's well known that acetaldehyde represents more than 90% of the total aldehyde content in wine. Its aroma threshold value is 100 mg L$^{-1}$; low levels of this compound give a desirable fruity aroma to the wines, whereas an excessive content produces an apple-like off-flavor in the wine, and levels more than 200 mg L$^{-1}$ cause wine flatness [51]. The highest amount of acetaldehyde was found in the wine produced in cellar P by the commercial strain (57.74 mg L$^{-1}$), although in all the wines, this compound did not exceed its threshold value.

Other compounds detected in this study were terpenes, which are responsible for wine varietal flavor, although they are not present at high levels in wine. It has been reported that besides grapes, yeasts are also involved in the production of terpenes. As shown in Table 6, four terpenes were

identified in these wines. The most represented was linalool, which has a rose-like floral aroma and contributes positively to wine aroma, although in these wines, it was detected at a value below its threshold value of 25 µg L$^{-1}$, and β-damascenone, which has been reported to possess sweet and exotic flavor notes. In addition, for almost all these compounds, wines produced by AWRI796 contained levels significantly different from the content detected in wines obtained with indigenous strains, particularly for wines produced in cellars B and M.

**Table 6.** Concentration (average ± SD) of volatile compounds detected in wines obtained in three cellars (B, P, and M) with commercial *S. cerevisiae* strain AWRI 796 (BC, PC, and MC, respectively) and indigenous *S. cerevisiae* strains isolated from each cellars (cellar B: BI; cellar P: PI; cellar M: MI). Values are expressed in µg L$^{-1}$, except those indicated with superscript letter "a" (mg L$^{-1}$). The asterisk (*) indicates differences statistically significant (Tukey's test, $p \leq 0.05$) between wines obtained with indigenous and commercial strains. The analysis was applied independently to wines obtained in each cellar.

| | Cellar B | | Cellar P | | Cellar M | |
|---|---|---|---|---|---|---|
| Compounds | BI | BC | PI | PC | MI | MC |
| **ESTERS** | | | | | | |
| Ethyl acetate [a] | 49.66 ± 0.17 * | 44.67 ± 0.44 | 17.89 ± 0.13 * | 19.28 ± 0.23 | 26.18 ± 0.27 * | 34.33 ± 0.37 |
| Ethyl propanoate | 151.24 ± 2.70 * | 125.18 ± 2.44 | 77.48 ± 1.43 * | 92.18 ± 1.57 | 147.16 ± 1.07 * | 166.54 ± 2.80 |
| Ethyl isobutyrate | 96.37 ± 3.07 | 107.71 ± 2.91 | 147.92 ± 4.20 * | 85.68 ± 1.10 | 70.42 ± 0.71 * | 103.34 ± 4.36 |
| Propyl acetate | 19.46 ± 1.85 | 26.18 ± 0.85 | 34.06 ± 1.35 * | 23.09 ± 0.99 | 59.71 ± 4.37 * | 44.82 ± 2.60 |
| Isobutyl acetate | 95.42 ± 2.77 | 82.85 ± 2.85 | 58.55 ± 1.75 | 68.82 ± 1.17 | 120.35 ± 15.27 * | 144.06 ± 3.68 |
| Isoamyl acetate | 429.17 ± 19.78 * | 345.19 ± 18.77 | 322.84 ± 1.53 | 369.84 ± 19.78 | 494.18 ± 12.78 * | 415.21 ± 23.91 |
| Hexyl acetate | 10.30 ± 25.25 * | 8.65 ± 0.18 | 4.34 ± 0.30 * | 6.16 ± 0.22 | 8.88 ± 0.55 * | 14.30 ± 0.44 |
| Ethylphenyl acetate | 9.25 ± 0.49 | 7.44 ± 0.99 | 3.86 ± 0.18 | 3.95 ± 0.23 | 5.77 ± 0.68 * | 8.96 ± 0.27 |
| 2-phenylethyl acetate | 55.06 ± 1.32 * | 42.86 ± 1.26b | 44.56 ± 0.82 * | 71.21 ± 2.71 | 21.24 ± 2.26 * | 65.62 ± 0.17 |
| Ethyl butyrate | 62.38 ± 1.13 * | 83.18 ± 2.91 | 62.40 ± 0.93 * | 52.72 ± 1.03 | 111.16 ± 1.39 * | 141.20 ± 2.97 |
| Ethyl hexanoate | 271.04 ± 2.39 * | 224.46 ± 3.58 | 162.96 ± 1.41 * | 211.10 ± 1.70 | 170.21 ± 3.88 * | 208.04 ± 9.14 |
| Ethyl octanoate | 25.25 ± 1.84 | 19.67 ± 2.38 | 8.73 ± 0.41 | 5.95 ± 0.38 | 17.52 ± 1.90 | 12.60 ± 0.85 |
| **ALCOLOLS** | | | | | | |
| 2-Propanol | 265.15 ± 1.38 * | 211.74 ± 7.38 | 160.69 ± 4.57 * | 141.31 ± 3.35 | 307.75 ± 4.47 | 320.30 ± 2.55 |
| 1-Butanol | 322.01 ± 1.27 * | 300.51 ± 3.11 | 182.21 ± 2.97 | 194.16 ± 2.84 | 407.05 ± 7.08 * | 374.51 ± 2.73 |
| 1-Hexanol | 90.33 ± 1.24 * | 72.43 ± 1.56 | 55.59 ± 0.85 * | 72.82 ± 1.53 | 112.65 ± 1.84 * | 149.74 ± 2.49 |
| Cis-3-Hexen-1-ol | 96.30 ± 0.59 * | 75.35 ± 1.58 | 64.11 ± 1.42 | 60.53 ± 1.69 | 167.26 ± 2.901 * | 133.96 ± 1.39 |
| Benzyl alcohol | 102.40 ± 1.44 * | 85.10 ± 1.18 | 101.55 ± 0.88 * | 94.82 ± 1.66 | 149.10 ± 0.91 * | 140.77 ± 1.68 |
| 2-phenylethanol | 379.22 ± 1.25 * | 297.14 ± 3.80 | 266.11 ± 2.40 * | 284.85 ± 1.48 | 448.18 ± 3.58 * | 550.12 ± 6.93 |
| N-propanol [a] | 48.32 ± 0.48 * | 60.38 ± 0.49 | 22.82 ± 0.21 * | 40.59 ± 0.28 | 27.15 ± 0.24 * | 49.19 ± 0.21 |
| Isobutanol [a] | 40.92 ± 0.62 * | 36.5 ± 0.54 | 28.41 ± 0.49 * | 24.87 ± 0.44 | 99.87 ± 0.23 * | 59.70 ± 0.42 |
| D-amyl alcohol [a] | 80.54 ± 0.57 * | 69.59 ± 0.57 | 86.70 ± 0.35 * | 78.32 ± 0.48 | 132.35 ± 1.20 * | 108.15 ± 1.65 |
| Isoamyl alcohol [a] | 219.17 ± 1.34 * | 72.10 ± 0.57 | 200.88 ± 1.20 * | 129.44 ± 0.62 | 299.87 ± 1.44 * | 246.75 ± 2.28 |
| **TERPENES** | | | | | | |
| Linalool | 13.22 ± 0.42 * | 9.78 ± 0.30 | 11.87 ± 0.29 * | 14.00 ± 0.15 | 11.22 ± 0.45 * | 15.44 ± 0.88 |
| α-Terpineol | 15.71 ± 0.28 * | 12.18 ± 1.07 | 8.12 ± 0.63 | 7.56 ± 0.45 | 10.58 ± 0.57 * | 15.64 ± 0.85 |
| β-Citronellol | 7.12 ± 0.48 | 6.56 ± 0.59 | 5.33 ± 0.32 | 6.11 ± 0.18 | 6.76 ± 0.38 * | 9.49 ± 0.57 |
| Geraniol | 8.64 ± 0.44 * | 6.72 ± 0.41 | 3.86 ± 0.17 | 4.63 ± 0.05 | 5.62 ± 0.39 * | 8.11 ± 0.30 |
| β-damascenone | 15.19 ± 0.61 * | 12.88 ± 0.31 | 8.93 ± 0.18 * | 11.75 ± 0.31 | 13.04 ± 0.36 | 14.57 ± 0.78 |
| **ALDEIDES** | | | | | | |
| Acetaldehyde [a] | 39.91 ± 0.16 * | 15.41 ± 0.38 | 17.60 ± 0.48 * | 57.74 ± 0.35 | 32.42 ± 0.38 * | 39.91 ± 0.28 |
| Benzaldehyde | 35.89 ± 1.27 * | 29.22 ± 1.25 | 14.56 ± 0.62 * | 23.74 ± 0.35 | 33.35 ± 0.90 * | 45.69 ± 0.57 |
| Furfural | 81.66 ± 1.27 * | 68.55 ± 0.57 | 60.49 ± 0.26 * | 49.43 ± 0.41 | 125.40 ± 2.21 * | 104.36 ± 2.29 |

All the parameters determined in wines obtained by indigenous and commercial strains in the three cellars were submitted to principal component analysis (PCA). The first two components account for about 79% of the total variance. The first principal component (PC1) explained 55.98% of data variability, and was correlated with ethyl propanoate, isobutanol, 2-propanol, 1-butanol, 2-phenylethyl acetate, and benzaldehyde, while propyl acetate, ethyl hexanoate, ethyl acetate, and *n*-propanol contribute more strongly to the second principal component (PC2). The plot of the six wines on the plane defined by these first two components is shown in Figure 3. The PCA of the wines revealed that the wines obtained in the same cellar by inoculating the two strains (indigenous and commercial) differed in the aromatic profile, as they were located in different quadrants. Only wines obtained in cellar P with both starters were located in the same quadrant. In both the fermentation trials performed

in cellar P, the starter dominance was the lowest one (Table 4), mainly for fermentation performed by the commercial starter. In this trial, a high participation of indigenous *S. cerevisiae* strains to fermentative process occurred, which might affect the composition, making the wine obtained by commercial starter similar to wine produced by inoculating the indigenous strain previously isolated and selected from the same grape must. This result confirms that dominance or competitiveness of a yeast starter strain could make a significant impact on the aromatic characteristic of wine by dominating its sensorial quality or eliminating the influence of the *S. cerevisiae* population naturally present in fermenting grape must [52]. Furthermore, the wines obtained inoculating the same starter, the commercial strain AWRI796, were located in three different quadrants, indicating differences in the chemical composition of wines obtained by using the same yeast strain, but in different wineries. Although the same grape variety was used in all the cellars, each of them were located in different geographical areas, which can be affected by the composition of grape must, i.e., the precursor content. In fact, factors that are also related to the vineyard growing area, such as seasonal weather differences, soil composition, and vineyard management were reported to affect the development and retention of grape aroma compounds, and consequently the aroma of the wines produced [53]. This result emphasizes that the effective impact of yeast strains on the aroma properties is dependent on a network of effects; other strain metabolisms and other factors, such as raw material composition and winemaking procedure, amongst others, have to be considered.

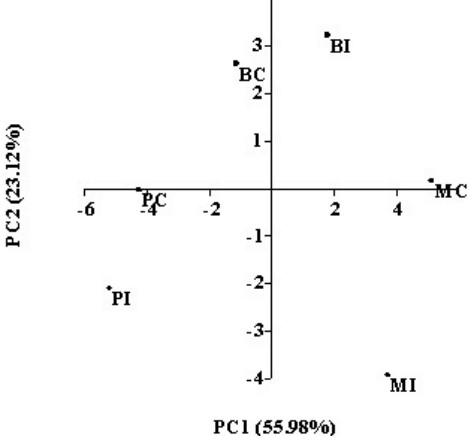

**Figure 3.** Principal component analysis (PCA) plot based on by-products detected in wines produced during pilot-scale vinification in cellars B, P, and M. BI and BC = wines obtained with indigenous and AWRI796 starters, respectively, in cellar B; PI and PC = wines obtained with indigenous and AWRI796 starters, respectively, in cellar P; MI and MC = wines obtained with indigenous and AWRI796 starters, respectively, in cellar M.

The wines obtained during pilot-scale vinifications were submitted to sensory evaluation in order to evaluate if there were significant differences on the overall organoleptic quality of wines obtained in the three cellars. As reported in Table 7, the results of a one-way ANOVA model performed on the sensory data showed a significant effect of samples on liking scores (F = 11.79; $p < 0.00$; L.S.D. = 0.49). The sample associated with the highest liking score was the wine obtained in cellar P by inoculating the commercial starter (sample PC), followed by wines obtained in the three different cellars by inoculating the indigenous starters (samples BI, MI, PI), whereas the lowest liking scores were associated with wines produced by inoculating the commercial starter in cellar B and M (samples BC and MC). By considering that a score of 5 was taken as the lower limit of acceptability, all the wines fermented with indigenous starters reached the threshold of acceptability, whereas among wines produced by the commercial strain, only the wine from cellar P attained a liking score much higher than the threshold of acceptability. It has to be underlined that during the fermentation of this wine, the inoculated starter showed a low dominance ability and a high participation of indigenous strains was found

(Table 4), which might affect both the aromatic and sensorial qualities of wine. However, these results indicate a predilection of the tasters toward the wines produced with selected indigenous starters or with a high contribution of indigenous *S. cerevisiae* strains, suggesting the need to support the use of indigenous starters for safeguarding the biodiversity of vineyard yeasts. It was found [54] that the winery environment and commercial strain use significantly alter the composition of *S. cerevisiae* population present in the first phase of spontaneous fermentation, indicating that a population of yeast descended from commercial strains might be resident in the winery facilities. As a consequence, the promotion of indigenous starter cultures represents a promising tool to limit the erosion of natural biodiversity induced by the wide use of commercial starters in winemaking.

**Table 7.** Liking scores of wines obtained in pilot-scale vinifications. BI and BC = wines obtained with indigenous and AWRI796 starters, respectively, in cellar B; PI and PC = wines obtained with indigenous and AWRI796 starters, respectively, in cellar P; MI and MC = wines obtained with indigenous and AWRI796 starters, respectively, in cellar M. Superscript letters indicate a significant difference between mean ratings ($p < 0.05$).

| Sample | Liking Scores |
|--------|---------------|
| PC | 6.38 [a] |
| BI | 5.53 [b] |
| MI | 5.12 [b,c] |
| PI | 5.03 [c] |
| BC | 4.97 [c] |
| MC | 4.66 [c] |

## 4. Conclusions

The results obtained in this study highlighted that biodiversity among wine yeasts collected directly from vineyard environments represents a high valuable source of yeasts possessing traits of technological interest. The *S. cerevisiae* strains tested in this study showed high variability both for technological and metabolic traits. It was found that the fermentation of the same grape must with different strains produces wines containing different amounts of aromatic compounds, but we have to consider that the metabolic activity is affected also by strain origin. These findings confirm that the aromatic quality of wine is the result of a strict interaction between grape must composition and yeast strains performing the fermentation. This is the result between yeast abilities to release volatile compounds from grape precursors and to synthesize de novo volatile compounds. At the end of this study, indigenous *S. cerevisiae* strains suitable to be used as starter culture were individuated, underlining the fundamental role of the screening program followed in the laboratory to select the most promising indigenous *S. cerevisiae* strains able to efficiently ferment specific musts. In fact, all the indigenous starters tested during pilot-scale vinification showed a dominance level higher than the commercial strain in all the cellars, confirming that indigenous yeast strains are better acclimated to the environmental conditions present in grape must to be fermented, being that these indigenous starters are highly competitive against yeast microflora that are naturally present in the grape must. This characteristic guarantees that the fermentative process is performed by inoculated starters, producing wine with expected characteristics. Furthermore, the wines obtained by inoculating these starters reached high liking scores during sensorial evaluation. The highest liking score was reached for wine obtained in cellar P with the commercial starter, but in this fermentation, a high contribution of indigenous strains was found, confirming the predilection of the tasters toward the wines produced with indigenous *S. cerevisiae* strains. This result confirms that indigenous strains are able to exalt the peculiarities of a wine, assuring the maintenance of the typical sensory properties of the wines of any given region, and this is particularly worrying when all winemakers in a particular region use a very limited number of commercial yeasts, which obviously results in the production of highly homogeneous wines. Employing these starters provides, other than technological advantages, an efficient strategy for winemakers to differentiate their products in a highly competitive market.

**Author Contributions:** All authors contributed to the conception and design of work; R.P., G.S., and R.R. performed the experiments; N.C. performed the sensory analysis; all the authors analyzed and elaborated the data; A.C. contributed to drafting and revising the work; P.R. contributed to revising the work, ensuring that questions related to the accuracy of any part of the work were appropriately investigated.

**Funding:** This work was supported by the project PSR Regione Basilicata 2014-2020 Sottomisura 16.1 GO Vite and VinoPROduttività e Sostenibilità in VITIvinicoltura—(PROSIT)—N. 54250365779.

**Conflicts of Interest:** The authors declare no conflict of interest.

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
