# Peer review of "Selected Indigenous Saccharomyces cerevisiae Strains as Profitable Strategy to Preserve Typical Traits of Primitivo Wine"

_fermentation, doi:10.3390/fermentation5040087_

Round 1

Reviewer 1 Report

The article entitled “Selected indigenous Saccharomyces cerevisiae strains as profitable strategy for winemakers to differentiate their products in a highly competitive market” by Angela Capece and colleagues provide an overview of high biodiversity of wine strains in comparison with a standard industrial strain. They clearly show the differences by metabolic analysis, which will provide deep insights into the readers participating fermentation. Therefore, I strongly recommend to accept it as a research paper after some minor corrections.

Minor comments:

Line 82, Please provide URL or references of the repository. Lines 90-91, Please provide cultivation time and starting OD. Line 105, Please provide company name and place, or composition of the BiGGY agar. Lines 109-110, Please briefly mention about the method so that the readers can understand solely by reading the manuscript. Line 141, Please indicate the speed by g or provide information about the rotator (company name, catalogue No., etc) so that the diameter is specified. Line 498, Please italicize "S. cerevisiae".

Author Response

Response to Reviewer 1 Comments

Point 1: Line 82, Please provide URL or references of the repository

Response 1: Our collection is not provided of URL. We have specified in the text.

Point 2: Lines 90-91, Please provide cultivation time and starting OD.

Response 2: We have added in the text the cultivation time and specified that we have used YPD agar medium.

Point 3: Line 105, Please provide company name and place, or composition of the BiGGY agar

Response 3: We have added in the text company name and place of the BiGGY agar.

Point 4: Lines 109-110, Please briefly mention about the method so that the readers can understand solely by reading the manuscript

Response 4: Ok, made.

Point 5: Line 141, Please indicate the speed by g or provide information about the rotator (company name, catalogue No., etc) so that the diameter is specified.

Response 5: We have indicated in the text the speed by g.

Point 6: Line 498, Please italicize "S. cerevisiae". 

Response 6: Ok, made.

Reviewer 2 Report

Review fermentation-593828

The title is not accurate regarding the content of the manuscript. In particular the title refers to market strategy although this is not the purpose of the manuscript. Primitivo cultivar should be indicated in the title.

The work was conducted straightforward and the results are clearly presented. However, the novelty of this kind of manuscript is limited, as based on the assumption that each wine producing region or cultivar would benefit from autochthonous yeast starters an almost infinite number of studies can be conducted. Moreover the results obtained do not clearly support the conclusion.

Some references could be added in the introduction or discussion sections, such as:

https://www.sciencedirect.com/science/article/pii/S221078431630002X

https://www.nature.com/articles/srep14233

https://mbio.asm.org/content/7/3/e00631-16.short

https://journals.plos.org/plosone/article?id=10.1371/journal.pone.0160259

This could broader the discussion as the selection of additional Saccharomyces starters is not

Moreover references 3, 17 and 44, in Italian or Spanish, are not easy to find and read for most of the audience and I suggest removing them.

L.117-118: indicate also assimilable nitrogen level.

Data from figure 1 could be included in Table 2, and letters for significance of differences added.

Table 3: one decimal for all values is more than enough.

Figure 2: please add vectors illustrating parameters.

Text and Table 4: the term implantation rate does not seem accurate. Use rather “proportion of starter” or something similar.

307-308: is that significant? The term implantation is relevant for the first stage of sampling. For the other stages, the observation is rather related to persistence. Moreover, with only 30 colonies assayed, the significance of differences should be specified.

Table 4: it would be more interesting to show identification profiles of isolates and their frequency as this is what is mainly commented in text.

340-342: most probably the difference is related to a feature which has not been analyzed, as ethanol level did not exceed 12%. In addition, the hypothesis is argued with data not shown, which is a weak evidence. 344 and following: Identification and counting of non-Saccharomyces yeasts is not described in the materials and methods section. Specify how many colonies were sampled and identified. This is crucial for interpreting the results. 382: High residual sugar levels observed for cellar P should be commented further.

Table 5: add a column with cellar name for clarity or use the same notation than in table 7 and figure 4.

498: italics for S. cerevisiae.

Conclusion is not clearly supported by the results. The authors should emphasize in discussion the three key elements which are reported in the discussion. As such the conclusion seems to rely on previously published data and not on data from this study.
